# Glacial lake inventory of High Mountain Asia in 1990 and 2018 derived from Landsat images

Xin Wang [1,2], Xiaoyu Guo [1], Chengde Yang [2], Qionghuan Liu [3], Junfeng Wei [1], Yong Zhang [1], Shiyin Liu [4], Yanlin Zhang [1], Zongli Jiang [1], Zhiguang Tang [1]

[1]School of Resource Environment and Safety Engineering, Hunan University of Science and Technology, Xiangtan, 411100, China

[2]State Key Laboratory of Cryospheric Science, Northwest Institute of Ecology and Environmental Resources, Chinese Academy of Sciences, Lanzhou 730000, China

[3]Key Laboratory of Land Surface Pattern and Simulation, Institute of Geographic Sciences and Natural Resources Research, Chinese Academy of Sciences, Beijing 100101, China

[4]Institute of International Rivers and Eco-security, Yunnan University, Kunming, 650000 China

*Correspondence to*: Xin Wang (xinwang_hn@163.com)

**Abstract.** There is currently no glacial lake inventory data set for the entire High Mountain Asia (HMA) area. The definition and classification of glacial lakes remain controversial, presenting certain obstacles to extensive utilization of glacial lake inventory data. This study integrated glacier inventory data and 668 Landsat TM/ETM+/OLI images, and adopted manual visual interpretation to extract glacial lake boundaries within a 10-km buffer from glacier extent using ArcGIS and ENVI software, normalized difference water index maps, and Google Earth images. The theoretical and methodological basis for all processing steps including glacial lake definition and classification, lake boundary delineation, and uncertainty assessment are discussed comprehensively in the paper. Moreover, detailed information regarding the coding, location, perimeter and area, area error, type, time phase, source image information, and sub-regions of the located lakes is presented. It was established that 27,205 and 30,121 glacial lakes (size: 0.0054–6.46 km$^2$) in HMA, covered a combined area of 1806.47 $\pm$ 2.11and 2080.12 $\pm$ 2.28 km$^2$ in 1990 and 2018, respectively. The data set is now available from the National Special Environment and Function of Observation and Research Stations Shared Service Platform (China): http://www.crensed.ac.cn/portal/metadata/706ce17f-1684-4e8d-bf5e-7d517e03693c

## 1 Introduction

Under the background of climate warming and the consequent widespread mass loss of glaciers in alpine regions, increasing volumes of glacial meltwater are being released. This results in glacial lake expansion and extended areas of low-lying terrain (e.g., depressions and troughs) left behind by

retreating glaciers in which water can accumulate and new glacial lakes can form (Clague and Evans, 2000; Mool et al., 2001; Song et al., 2016). As both a water resource and a source of flash flood/debris flow hazards, glacial lakes participate in several natural processes, e.g., regional energy and water cycles (Slemmons et al., 2013), act as both indicators and containers of environmental information (Wang et al., 2016, 2019b; Zhang et al., 2019), and drive hillslope erosion and landscape evolution (Cook et al., 2018) in the alpine cryosphere. On the one hand, glacial lakes act as temporary storage for the meltwater resource because a considerable amount of meltwater is retained by glacial lake expansion, e.g., approximately 0.2 % $a^{-1}$ of the total glacial meltwater was reserved in glacial lakes from 1990 to 2010 in the Tien Shan Mountains in Central Asia (Wang et al., 2013). On the other hand, given the worldwide expansion in lake area in recent decades, the potential will increase for glacial lakes to develop into glacial lake outburst floods and related debris flows that could threaten downstream residents, infrastructure, and regional ecological and environmental security (Huggel et al., 2002; ICMOD, 2011; Bolch et al., 2012; Haebeli et al., 2016). Thus, glacial lakes perform important roles both in the meltwater cycle and in glacier hazard evolution in the cryosphere.

Following the rapid development of remote sensing technology and computer science, remote sensing imagery acquired by various satellites and sensors has been used widely in glacial lake research. In particular, Landsat imagery has become the most important data source for dynamic investigation of glacial lakes because of its wide coverage, continuous and long-term temporal sequence, and accessibility. Based on remote sensing data, both the distribution and the characteristics of change of glacial lakes in the mountains and watersheds in the High Mountain Asia (HMA) region have been widely reported (Supplementary Table S1). For example, multi-source remote sensing imagery has been used to compile glacial lake inventories for regions of the Tibetan Plateau (Zhang et al., 2015), Tien Shan Mountains (Wang et al., 2013), Himalaya (Gardelle et al., 2011; Nie et al., 2017), Hengduan Mountains (Wang et al., 2017), Uzbekistan (Petrov et al., 2017), Pakistan (Senese et al., 2018), and HMA, excluding Altai and Sayan (Chen et al., 2020). These inventories have proved an important data resource both for revealing the spatiotemporal characteristics of glacial lakes and for understanding the response of glacial lakes to the effects of climate change in these regions.

Automatic and semi-automatic glacial lake boundary vectorization approaches have been used most widely in regional glacial lake investigations because of their higher efficiency and objectivity in comparison with manual visual vectorization. In such research, water bodies are usually determined based on the characteristics of different remote sensing bands and computer-dependent algorithms, e.g.,

the normalized difference water index (NDWI), band ratio, support vector machine, decision tree, spectral transformation, object-oriented classification, global–local iterative scheme, active contour model, and random forest (Gardelle et al., 2011, Huggel et al., 2002, Li et al., 2011; Veh et al., 2018, Zhang et al., 2018). However, manual post-processing is often required to calibrate the uncertainties that could easily be produced by the above approaches. Furthermore, the labour costs associated with rectification of lake boundary errors increase sharply with increasing complexity of study area terrain (Yang et al., 2019). With consideration of the accuracy, efficiency, and time overheads associated with the various vectorization approaches, a manual vectorization approach was adopted for investigation of the glacial lakes on the Tibetan Plateau (Zhang et al., 2015) despite the labour requirements and the anticipated additional errors produced by individual subjectivity (Nie et al., 2017; Yang et al., 2019; Song et al., 2014).

Controversies and knowledge gaps remain regarding available glacial lake inventories for different alpine cryosphere regions, which present certain obstacles to extensive utilization of glacial lake inventory data. The main problems relate to regional differences in lake development, inconsistent specifications of lake definition, and the adoption of various approaches regarding lake interpretation (Yao et al., 2018). There is no existing comprehensive glacial lake inventory for the entire HMA and knowledge regarding the spatiotemporal characteristics of glacial lakes in this region remains incomplete. The objectives of this study were to fill this knowledge gap by producing a glacial lake inventory data set for HMA derived from Landsat images, and to provide fundamental data for water resource evaluation, assessment of glacial lake outburst floods, and glacier hydrology research in the mountain cryosphere region.

**2 Study area**

The HMA area mainly comprises the Tibetan Plateau and surrounding alpine ranges. The area is divided into 13 sub-regions in version 5 of the Randolph Glacier Inventory (RGI 6.0), i.e., the Himalaya area (Western Himalaya, Central Himalaya, and Eastern Himalaya), Hengduan Mountains, Southern and Eastern Tibet, Inner Tibetan Plateau, Karakoram and Western Kun Lun, Qilian Shan and Eastern Kun Lun, Hindu Kush, Pamir, Alay and Western Tien Shan, Eastern Tien Shan, and Altay and Sayan (Arendt et al., 2015; Pfeffer et al., 2014). The boundaries of the 13 sub-regions and outlines of the glaciers in HMA derived from RGI 6.0 are shown in Figure 1. This region (26°–54°N, 67°–104°E) is characterized by tremendously complex topographic conditions with widespread distribution of mountain glaciers. According to the Climatic Research Unit Time Series v4.02 data set

(http://data.ceda.ac.uk/badc/cru/data/cru_ts/cru_ts_4.02/), the air temperature of the different
sub-regions in HMA increased at an average annual rate of 0.002–0.054 °C a⁻¹ during 1990–2018 (Fig.
1). The annual rate of change of precipitation in HMA during 1990–2018 varied from −9.9 to 4.2 mm
a⁻¹ with a small average rate of increase of 0.3 mm a⁻¹.

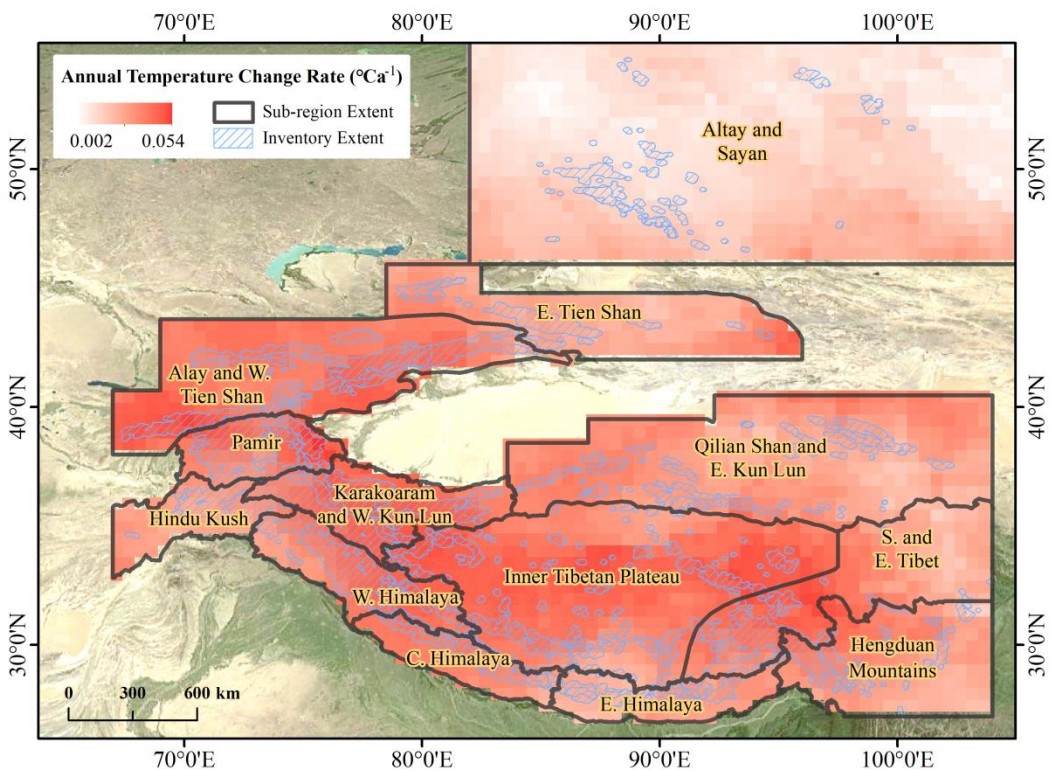

Figure 1. Location of sub-regions, rate of change of air temperature (1990–2018), and buffer area within
10 km of glacier extent for glacial lake inventory of High Mountain Asia.
The HMA area has the largest surviving glaciers of any region other than the polar regions. As
reported in RGI 6.0, there were 97,974 modern glaciers in our study area, covering a total area of
approximately 98,768.86 km2. Together, these glaciers produced an average negative mass balance of
−150 ± 110 kg m−2 a−1 (Hock, et al., 2019 ) , which was the primary source of water supply for the
development of glacial lakes. Over recent decades, glaciers in most areas of HMA appear to have
experienced widespread mass wastage and area shrinkage (Bolch et al., 2012; Yao et al., 2012; Kääb et

al., 2012; Brun et al., 2017). However, the so-called "Karakoram Anomaly" refers to a region that is a prominent exception, which is characterized by glaciers with stable or positive mass balance (Hewitt, 2005; Gardner et al., 2013; Kääb et al., 2015).

**3 Data source**

We developed our glacial lake inventory of HMA based on 668 high-quality images selected from more than 1800 Landsat images with 30-m spatial resolution derived from the websites of the United States Geological Survey (https://www.usgs.gov/) and Geospatial Data Cloud (http://www.gscloud.cn/). To ensure the accuracy of glacial lake boundary extraction, the following criteria were applied to imagery selection. First, the cloud coverage in an image had to be <10 %. Second, for areas with no eligible or only low-quality imagery (because of snow or shadows) in the given year, acceptable images from years closest to the given year were chosen as replacements (Fig. 2). Third, images acquired in summer or autumn (June–November), when lake areas were believed near or at their maximal extent, were set as optimal choices to minimize the impact produced by seasonal area changes of the glacial lakes (Fig. 2). Based on the above criteria, 394 and 274 Landsat images were selected to represent circa 1990 and circa 2018, respectively, which completely covered the buffer area within 10 km of glacier extent acquired from the Second Chinese Glacier Inventory (http://westdc.westgis.ac.cn) and RGI 6.0 (https://www.glims.org/RGI/rgi60_dl.html). Among the selected images, those acquired during summer and autumn (June–November) accounted for 82.0 % of the total number of selected images, while those acquired during autumn (September–November) accounted for 56.9 % of the total number. In addition, a Shuttle Radar Topography Mission digital elevation model with spatial resolution of 1″ (http://imagico.de/map/demsearch.php) was used to derive the elevation of the glacial lakes.

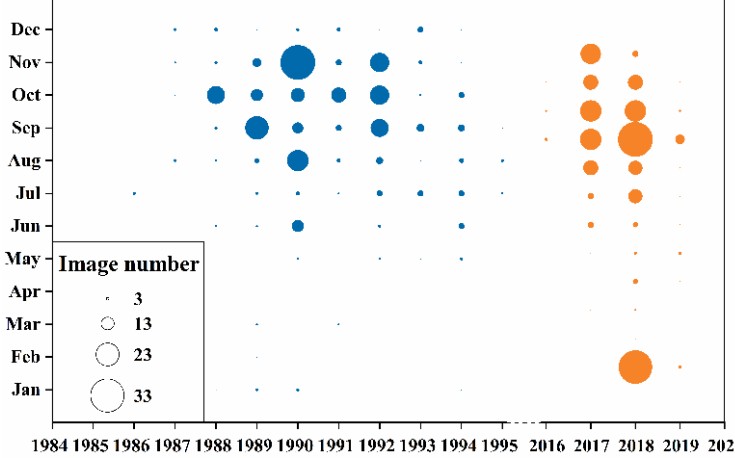

Figure 2. Time phases of remote sensing images selected for High Mountain Asia glacial lake inventory.
**4 Glacial lake inventory methods**
**4.1 Outline of workflow**
The methods and workflow adopted in this study to produce the glacial lake inventory mainly
included collation of knowledge and formulation of the specifications of the glacial lake inventory, data
pre-processing, manual vectorization of glacial lakes, interactive checking and error controlling, and
attribute database assignment (Fig. 3).
(1) Collation of available knowledge regarding glacial lake inventories. As much literature as
possible relevant to the investigation and recording of glacial lakes was collected. The various
definitions and classifications of glacial lakes, as well as the methods adopted previously for glacial
lake boundary extraction and assessment of the extent of glacial lake distributions, were summarized
and normative rules formulated for the HMA glacial lake inventory, as explained further in Sect. 4.2.
(2) Formulation of the specifications of lake identification. First, a working group of four leading
experts in the field was founded in 2014 to discuss and formulate the specifications of the glacial lake
inventory. Current knowledge regarding identification of lakes from Landsat imagery (e.g., pixel colour,
lake shape, and lake background features) and specifications of vectorization (e.g., viewing scale of
1:10,000 on a computer screen vectorization of mixed pixels) was discussed and unified operating
criteria were compiled to guide the glacial lake inventory operatives. Novice vectorization operatives
were trained until their vectorization results met the pre-specifications of the inventory.
(3) Pre-processing of remote sensing data. Pre-processing of the Landsat imagery included false
colour compositing and calculation of NDWI maps. The false colour composite images were based on
combinations of the operating bands of 7, 5, and 2 or 4, 3, and 2 for Landsat TM/ETM+ images and 5,
4, and 3 for Landsat OLI images. The preliminary lake extent was extracted automatically from each
image over the entire HMA area using the NDWI based on the near infrared band (NIR) and green
band (GREEN), which represent the minimum and maximum water reflectance, respectively
(McFeeters, 1996; Zhai et al., 2015; Li et al., 2016; Zhang et al., 2018):
$$\qquad NDWI = \frac{B_{GREEN} - B_{NIR}}{B_{GREEN} + B_{NIR}} \qquad\qquad (1)$$
where $B_i$ is the spectral band of Landsat imagery. The NDWI maps were calculated for each selected
Landsat image using different region-specific thresholds. Liu et al. (2016) and Du et al. (2014)
suggested that it might be preferable to set the optimized threshold of $NDWI_{GREEN/NIR}$ of Landsat OLI
images to −0.05. By considering the edge effects according to the mixed pixels, this study initially
selected a lower optimal threshold (approx. −0.1) for specific images to obtain the maximum water
body. Then, higher thresholds were tested for visual water extraction before a suitable threshold (varied
in the range of −0.10 to 0.20) was selected for a given image to obtain the NDWI map. When manual
vectorization was performed on a false colour composite image, the NDWI maps of potential glacial
lakes were overlaid to assist in glacial lake identification.
(4) Manual vectorization and entering of attribute data. The inventory work was performed during
2014–2019. Seven groups were formed to conduct lake boundary vectorization of the 13 HMA
sub-regions. After vectorization of a glacial lake, it was required that manual attribute items (e.g., data
source and lake type) be input concurrently.
(5) Interactive checking and accuracy control. First, glacial lakes were discerned via human–
computer interaction, i.e., potential glacial lakes were revealed by the NDWI maps or identified
visually from the false colour composite images. Second, glacial lake boundary vectorization results
were checked interactively by another vectorization operative to eliminate misclassified areas of
shadow and ice and to add areas of glacial lakes evidently omitted in the boundary extraction process.
This checking process also minimized the subjective judgment errors of the operatives. Third, attribute
items such as glacial lake classification, new/disappeared lakes, and separated/coalesced lakes were
checked interactively. In this process, Google Earth imagery was used as an important auxiliary
reference data source for error examination.

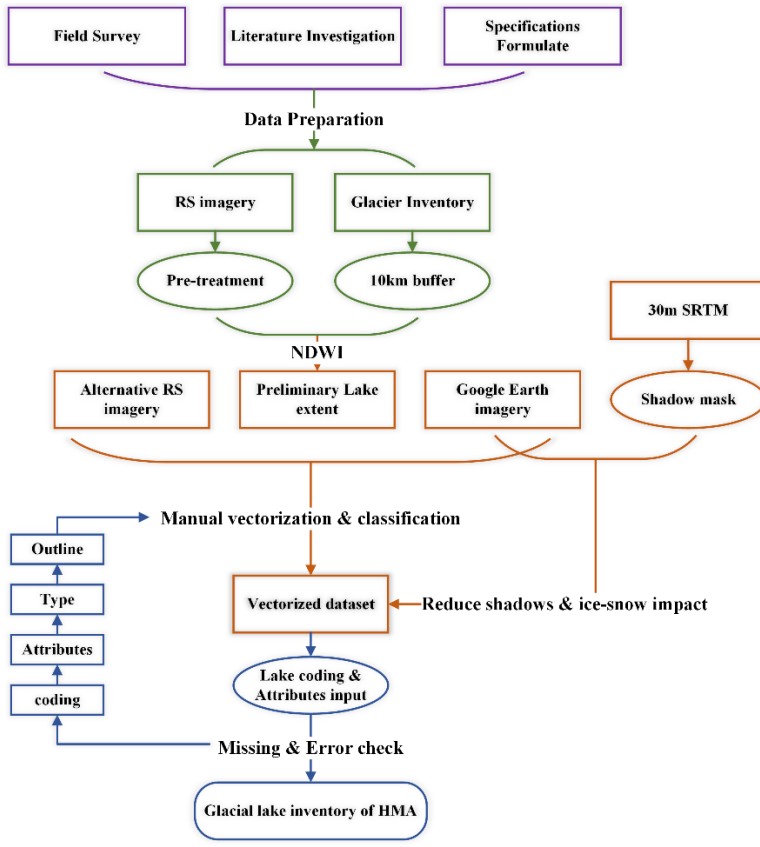

Figure 3. Flow chart of HMA glacial lake inventory.
**4.2 Illustration of key methods**
**4.2.1 Definition of glacial lakes**
The definition of a glacial lake determines the type of cryosphere water body that will be recorded
as a glacial lake. There are multiple definitions of a glacial lake based on different perspectives (Mool
et al., 2001; Yao et al., 2018). When glacial lake inventories are undertaken, most emphasize the
elementary role of glaciation in the formation of glacial lakes (Clague and Evans, 2000; Mool et al.,
2001). The remarkable difference is whether the period of glaciation or the supply source of glacial
lakes is given greatest attention. Some studies that focused on the former proposed that a glacial lake is
a natural water body formed by alpine glacier movement since the Last Glacial Maximum, i.e., ancient
or modern glaciers (Liu et al., 1988; Costa and Schuster, 1988). However, other studies emphasized the
relation of glacial lakes to meltwater in glaciated areas (Wang et al., 2013; Wang et al., 2014; Zhang et
al., 2015). The glacial lake inventory data compiled in this study are intended for use both in water
source evaluation and in assessment of environmental change in the alpine cryosphere. Thus, lakes
related to glaciers or to glaciation in the alpine cryosphere were all recorded as glacial lakes.
Most Quaternary glaciers have disappeared and the remaining relics are incomplete, which makes
it difficult to recover a continuous and complete glaciation range in alpine regions. Thus, it is of great
importance to ensure the range of glaciation in an alpine region when conducting a glacial lake
inventory based on remote sensing data. The most practical approach might be to specify an indicator
threshold to define the glaciation extent according to relevant findings of existing glacier relics in a
typical region. On the one hand, the glaciation frontier can usually be indicated by a specified lowest
elevation threshold, which is generally closely related to the regional climatic context caused by the
elevation effect. However, the lowest elevation threshold might vary enormously with respect to
different regions because regional climatic settings differ. For instance, the lowest elevations of 1700 m
in Austria (Buckel et al., 2018), 2000 m in Pakistan (Senese et al., 2018), 3000 m in Nepal and Bhutan
(Mool et al., 2001), and 3500 m in Peru (Hanshaw and Bookhagen, 2014) were used as specified
elevation thresholds to record glacial lakes. On the other hand, defining glaciation extent within a
specific distance from modern glacier terminals could be more suitable for the establishment of a
glacial lake inventory in relatively large-scale regions with complex regional climate, because the
differing climate within large-scale regions can be indicated approximately by the lowest elevation of
individual glacier terminals. Some studies adopted distances of 2, 3, or 10 km from modern glacier
terminals as thresholds with which to define areas of glacial lakes (Petrov et al., 2017; Veh et al., 2018;
Wang et al., 2012, 2013). Distances of 2, 5, 10, and 20 km were considered by Zhang et al. (2015). They
found that a distance of 10 km from a modern glacier terminal might be a reasonable guide to glaciation
extent and a threshold suitable for a glacial lake inventory of the Tibetan Plateau. This was supported by
the finding that the most distant glacierized boundary of the Little Ice Age was up to 10 km from the
modern glaciers in the Himalaya area (Wang et al., 2012, Nie et al., 2017). Additionally, to record
glacial lakes more precisely, combined distance and elevation thresholds have been used simultaneously
to define areas of glacial lakes in special small regions, e.g., lakes at elevations above 1500 m and within
2 km of modern glaciers were recorded as glacial lakes in Uzbekistan (Petrov et al., 2017). In this study,
given the large scale of the HMA region with its complex climatic context and extremely varied terrain,
the data set compiled included glacial lakes within a buffer zone of 10 km from modern glacier extent,
which covered an area of approximately $1.25 \times 10^6 \, km^2$ according to the Second Glacier Inventory of
China and RGI 6.0 (Fig. 1).
**4.2.2 Classification of glacial lakes**
In glaciation regions, the characteristics of glacial lakes, which include the phase of lake formation,
lake basin topography, dam material constituents, geometrical relationship with modern glaciers, and
source of water supply (or combinations thereof), have been employed as the basis for glacial lake
classification systems (Huggel et al., 2002; Liu et al., 1988; Mool et al., 2001; Yao et al., 2018). For
instance, based on lake basin topography, lakes in an inventory of the Hindu Kush–Himalaya region
were classified as erosion lakes, valley trough lakes, cirque lakes, blocked lakes, lateral and end
moraine-dammed lakes, and supraglacial lakes ( Liu et al., 1988; Mool et al., 2001). Recently, Yao et al.
(2018) presented a complete classification schema for glacial lake inventory and study of glacial lake
hazards that included six classes and eight sub-classes based mainly on the mechanism of glacial lake
formation, lake basin topography, and the geometrical relationship with modern glaciers.
Generally, it is a little difficult to distinguish glacial lake type in terms of material properties,
topographic features, and phase of lake formation using remote sensing imagery. Moreover, most of the
standards mentioned above were found inapplicable in previous studies of glacial lake classification in
large-scale regions such as HMA because of the lack of enough remote sensing data with satisfactory
spatial resolution. In this study, the hydrologic relationship between glacial lakes and modern glaciers
was adopted as a classification criterion because the present data set is intended to provide fundamental
data for water resource evaluation and glacier hazard assessment. Consequently, glacial lakes were
divided into just two types: glacier-fed lakes and non-glacier-fed lakes. The glacier-fed lakes were
further divided into three sub-classes: supraglacial lakes (lakes developed on glacier surface),
ice-contacted lakes (lakes contacting the glacier terminal or margin), and ice-uncontacted lakes (lakes
not contacting the glacier but fed directly by glacial meltwater). This classification was based on
whether the surface hydrological flow of the modern glacier and topographic features of the lake basin
allowed a lake to receive meltwater from the modern glacier. To achieve reliable classification results,
glacial lakes were distinguished with the assistance of 3D digital terrain imagery from Google Earth, a
Shuttle Radar Topography Mission digital elevation model, and glacier outlines from RGI 6.0. Based
on visual inspection of the satellite images and with reference to 3D digital terrain imagery from
Google Earth, we recorded a glacier-fed lake when (1) a lake had lower elevation than the modern
glacier (mother glacier) and (2) the mother glacier(s) meltwater could flow into the lake via surface
channel. It is common for glacial lakes to be fed by meltwater through subsurface channels; however,
we ignored this because it is difficult to survey the subsurface channels of glacial lakes using remote
sensing data. In addition, lake type was distinguished based on the topographic features of the lake
basin and the modern glaciers; in most cases, it is possible that the lakes were fed by meltwater flowing
through both sub-surface and surface channels.
**4.2.3 Extraction of lake boundary**
This study adopted automatic glacial lake extraction and manual glacial lake boundary
vectorization to determine glacial lake boundaries. In the NDWI-based automatic lake boundary
extraction approach, two bands were selected to facilitate a ratio calculation to maximize the difference
between water and non-water objects in the remote sensing imagery based on a given threshold. The
given threshold was determined subjectively with consideration of how much detailed information of
the lake water bodies was captured precisely. The given threshold was varied to account for various
factors such as the differences in Landsat sensors (i.e., TM, ETM+, and OLI), time phase of images,
quality of images, and complexity of surface features. To achieve the optimal threshold for lake water
body recognition, the candidate threshold was debugged iteratively for each image. In practice, because
the area of the glacial lakes was usually small (see next paragraph) and the spectral features of the lake
water bodies were varied, the threshold had to be set to allow capture of the greatest number of water
body pixels, which consequently resulted in simultaneous acquisition of more non-lake-water-body
noise information. It also resulted in more effort in the subsequent manual modification to reduce noise
information using methods such as algorithms to eliminate mountain shadows (Gardelle et al., 2011).
Manual visual vectorization distinguishes lake boundaries by identifying the unique texture,
colour, and other characteristics of glacial lakes in false colour composite images based on available
professional knowledge and accumulated experience in vectorization operations. Even though it was
regarded a time-consuming and labour-intensive process, it was also considered an attractive approach
because of its consistency, high level of quality control, and reasonably simple operational procedure,
given the varied quality of Landsat images available for the large-scale HMA region. In this study, the
manual visual vectorization process was generally found more suitable in terms of effort and precision
for generating a glacial lake inventory data set of the HMA region in comparison with automatic glacial
lake extraction. Therefore, manual visual vectorization in conjunction with NDWI maps was the main
method adopted to extract glacial lake boundaries to minimize the deficiencies produced by individual
subjectivity of the operatives.
The minimum number of pixels used to extract a glacial lake water body was found inconsistent in
the available literature. For example, arbitrary threshold areas of 0.0027 km$^2$ (three lake water body
pixels) (Zhang et al., 2015) and 0.0081 km$^2$ (nine lake water body pixels) (Nie et al., 2017) have been
used in earlier glacial lake investigations. Moreover, minimum threshold areas of 0.01 km$^2$
(approximately 10 lake water body pixels), 0.02 km$^2$ (approximately 22 lake water body pixels), and
0.1km$^2$ (approximately 111 lake water body pixels) have also been set to evaluate the level of risk of
glacial lake outburst floods in the Himalaya and Tien Shan Mountains (Petrov et al., 2017, Wang et al.,
2013; Worni et al., 2013; Bolch et al., 2011; Allen et al., 2019). Theoretically, one pure pixel of a lake
water body could be recorded as a glacial lake. However, a glacial lake is generally not represented by
one pure pixel unless it is aligned perfectly with the raster grid; usually, it would be surrounded partly
or fully by 1–8 mixed lake water body pixels (Fig. 4a, b). Consequently, manual delineation was
required for approximately 1/2, 1/8, or 7/8 of the peripheral mixed pixels surrounding pure lake water
body pixels (Fig. 4 d, e). If 3 or 4 pure lake water body pixels exist in a Landsat image, the maximum
number of peripheral mixed pixels is 12 (Fig. 4d, e). Usually, for three pure lake water body pixels, the
ratio of the area of pure lake water body pixels to the area of peripheral mixed pixels can be expressed
as follows:
$$\frac{\text{Area of peripheral mixed pixels}}{\text{Area of three pure lake water body pixels}} \times 100\% = \frac{6 \times \frac{1}{2} + 5 \times \frac{1}{8} + 1 \times \frac{7}{8}}{3} \times 100\% = 150\%. \qquad (2)$$
For four pure lake water body pixels, the ratio of the area of pure lake water body pixels to the area of
peripheral mixed pixels is:
$$\frac{\text{Area of peripheral mixed pixels}}{\text{Area of four pure lake water body pixels}} \times 100\% = \frac{8 \times \frac{1}{2} + 4 \times \frac{1}{8}}{4} \times 100\% = 112.5\%. \qquad (3)$$
Thus, in this study, the minimum glacial lake area recorded was set at 0.0054 km2 (e.g., 3–4 pure lake
water body pixels with approximately 12 peripheral mixed pixels, which equate to approximately 6 full
lake water body pixels) because a lake area covering fewer than three pure lake water pixels could
possibly have an error of >100 % (Fig. 4b, –c) despite the revised coefficient of one standard deviation
(0.6872) involved (see Sect. 5).

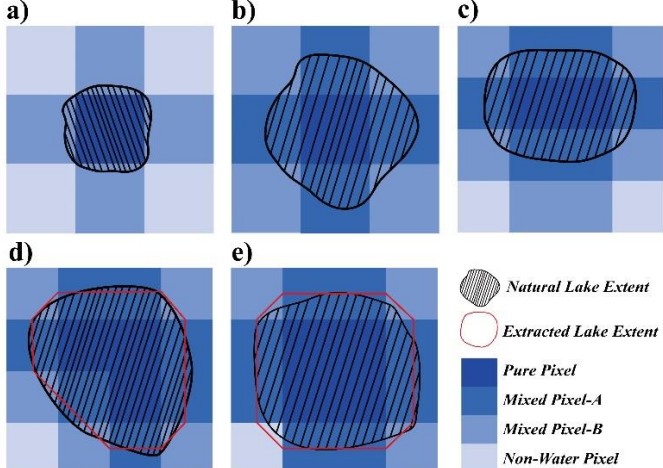

Figure 4. Sketches showing the relationships of pure water body pixels and surrounding potential
mixed water body pixels: (a) a pure water body pixel surrounded by four potential mixed water body
pixels, (b) a pure water body pixel surrounded by eight potential mixed water body pixels, (c) 2 pure
water body pixels surrounded by 10 potential mixed water body pixels, (d) 3 pure water body pixels
with 12 surrounding potential mixed water body pixels, and (e) 4 pure water body pixels with 12

8                            surrounding potential mixed water body pixels.

**4.2.4 Input of attribute items**
Eight attribute items were input into the HMA glacial lake inventory: lake coding, location
(longitude, latitude, and elevation), perimeter, area, type, area error, time phase, source image
information, and sub-region of located lake. (1) We encoded each glacial lake based on its central
location using the same coding format as used by the National Snow and Ice Data Centre to encode
glaciers. The code can be expressed as "GLmmmmmmEnnnnnN", where m and n represent the results
of the longitude and latitude of each glacial lake centroid multiplied by 1000, respectively, GL is the
abbreviation of glacial lake, and E and N represent eastings and northings, respectively. (2) The
location information of each glacial lake was labelled as the geographic coordinates of the centroid of
the shape of each glacial lake, calculated using ArcMap software. The lake elevation was defined as the
average elevation of a buffer zone of 30 m radius centred on the glacial lake centroid, which was
derived from the Shuttle Radar Topography Mission digital elevation model. (3) The area and
perimeter of each lake were calculated using ArcMap based on the unified geography coordinate
system of GCS_WGS_1984 and the Asia_North_Albers_Equal_Area_Conic projection system,
respectively, to avoid errors caused by projection deformation. (4) The error of lake area was calculated
using Eqs. (4) and (5) (Sect. 5). (5) Lake type, which was input manually, was defined as either
supraglacial lake, ice-contacted lake, ice-uncontacted lake, or non-glacier-fed lake (see Sect. 4.2.2). (6)
Lake time phase was the acquisition date of the original Landsat image, which was recorded as the time
phase for each lake. (7) Source image information referred to the image number of the Landsat images
used to extract the glacial lake boundary. (8) The sub-region to which each lake belonged identified the
regional location within the HMA area. Each lake was assigned based on shp. file data of the
boundaries of the 13 HMA sub-regions, obtained from the National Snow and Ice Data Centre, using
the ArcMap spatial analysis tool.
**5 Error assessment**
The errors associated with glacial lake extraction from remote sensing imagery using manual visual
delineation are generally related to components of the quality of the images (e.g., spatiotemporal
resolution, cloud coverage, and mountain shadows), experience, operative subjectivity, and the threshold
area of the inventory (Gardelle et al., 2011; Hall et al., 2003; Paul et al., 2004; Salerno et al., 2012; Zhang
et al., 2015). It has been reported that the area error of glacial lake boundary extraction based on remote
sensing images can be approximately ±0.5 pixels depending on the quality of the imagery (Fujita et al.,
2009; Salerno et al., 2012). Furthermore, the area error of glacial lake delineation attributable to manual
delineation can be assumed to follow a Gaussian distribution (Hanshaw and Bookhagen, 2014). Hence,
the theoretical maximum area error of glacial lake boundary extraction is the half-area of the edge pixels
because pure lake water body pixels are usually surrounded by mixed pixels (Fig. 4). The lake area
error of a single glacial lake within one standard deviation (1σ) can be expressed as follows (Hanshaw
and Bookhagen, 2014):
$$\text{Error}(1\sigma) = \frac{P}{G} \times \frac{G^2}{2} \times 0.6872, \tag{4}$$
$$\text{E} = \frac{\text{Error}(1\sigma)}{A} \times 100\ \% \ , \tag{5}$$
where $P$ is the perimeter of the glacial lake (m), $G$ is the spatial resolution of the remote sensing
imagery (30 m in this data set), 0.6872 is the revised coefficient under 1σ (i.e., approximately 69 % of
peripheral pixels are subjected to errors), $E$ is the relative error of the glacial lake, and $A$ is the total

area of the glacial lake. Then the accumulation of errors of all the lake areas for the entire study region or subregions can be calculated using the following formula based on error propagation theory:

$$E_T = \sqrt{\sum_{i=1}^{n} a_i^2} \,, \tag{6}$$

where $E_T$ is the area error of the entire study region or subregions, *i* is the lake of No. *i* in the entire study region or sub-regions, and *a* is the error area of a single lake.

The resulting calculated error indicated that the total absolute area error of HMA glacial lakes was approximately $\pm 2.11$ and $\pm 2.28$ km$^2$ and the average relative error was $\pm 13.5$ and $\pm 13.2$ % in 1990 and 2018, respectively. The relative area errors of each lake varied from 1–79 %, and a significant power exponential relationship was found between the relative area error and the sizes of the glacial lakes ($E = 0.050A^{-0.45}$, $R^2 = 0.96, \alpha < 0.001$) (Fig. 5a). Small-sized lakes (i.e., area $\leq 0.01$ km$^2$, which accounted for 2 % of the total lake area in HMA) had the largest average relative area error of 44.6 % (Fig. 5b). Medium-sized lakes (i.e., area of 0.01–0.1 km$^2$, which accounted for 34 % of the total lake area in HMA) had an average relative area error of 22.0% (Fig. 5c). Large-sized lakes (i.e., area $\geq 0.1$ km$^2$, which accounted for 64 % of the total lake area in HMA) had the smallest average relative area error of 7.6% (Fig. 5d). In summary, smaller glacial lakes in the HMA region had larger relative area errors, and vice versa.

To further verify the accuracy of the manual delineation of glacial lake boundaries, nine lakes located within the HMA region were surveyed using a portable GPS device (Trimble GeoXH6000) with decimetre accuracy during July–August 2018 (Fig. 6). The lakes selected for field survey covered areas of 0.01–2.97 km$^2$. The field-based lake boundaries were compared with those obtained via manual delineation (i.e., derived from Landsat OLI imagery acquired during 2018). It was found that the area error (i.e., the percentage difference of the absolute area encircled by the manually delineated lake boundary and that derived by the GPS survey) varied from 5.5–25.5 %. Moreover, it was determined that the average horizontal distance deviation between the two types of boundary varied from 4.5–33.5 m (Table 1). Overall, the horizontal deviations were largely confined to one pixel, and the average accuracy of the delineation of glacial lake boundaries was within $\pm 0.5$ pixels ($\pm 15$ m).

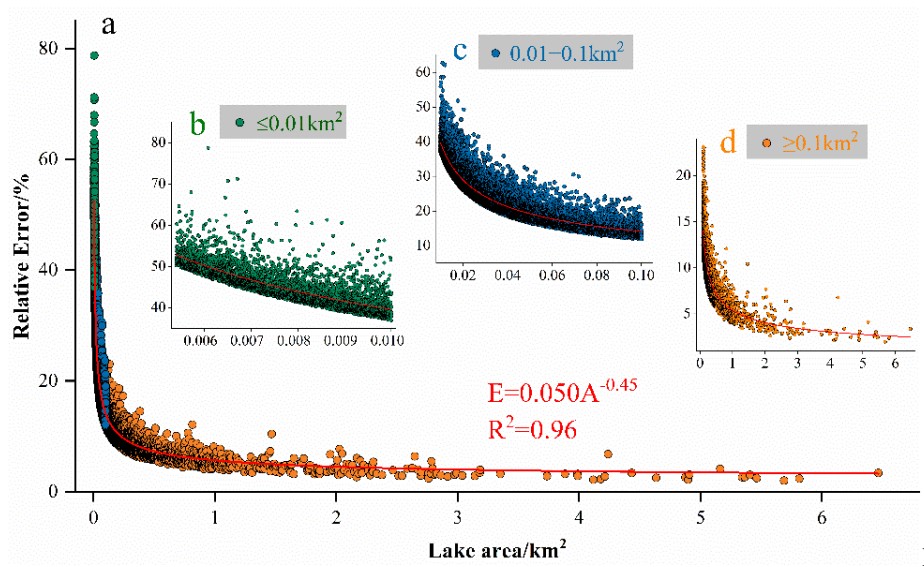

1                                  Figure 5.

2     Relationships of relative area error against size of glacial lakes in HMA: (a) relationship for glacial

3             lakes of all sizes and (b)–(d) relationships for glacial lakes of specific size.

5     Table 1. Horizontal deviations between lake boundaries obtained by manual delineation and field

6             survey using a portable GPS device (Trimble GeoXH6000)

| Name (labelled in Figure 6 ) | Lake ID | Lake size (km$^2$) | Horizontal deviations of delineation boundary (m) | | | Area error(%) |
|---|---|---|---|---|---|---|
| | | | minimum | maximum | average | |
| Qiongyong Cuo (A) | GL090225E28890N | 0.08 | -7.6 | 9.1 | 4.5 | 5.5 |
| Passu Lake (B) | GL074878E36457N | 0.15 | -10.9 | 12.0 | 6.0 | 6.8 |
| Longbasa Lake (C) | GL088071E27950N | 1.49 | -22.7 | -8.4 | 12.4 | / |
| Zongge Cuo (D) | GL087654E28113N | 1.48 | -26.8 | 24.9 | 13.5 | 6.1 |
| Unnamed (E) | GL088151E28010N | 0.01 | -4.7 | 4.8 | 3.2 | 16.3 |
| Unnamed (F) | GL088257E28011N | 0.58 | -12.9 | 12.4 | 4.6 | / |
| Unnamed (G) | GL088240E28005N | 0.40 | -20.8 | 15.9 | 7.1 | / |
| Large Laigu Lake (H) | GL096818E29298N | 2.97 | -36.7 | -6.4 | 15.3 | / |
| Small laigu Lake (I) | GL096832E29294N | 1.02 | -32.8 | 17.6 | 9.8 | / |

7    Note: "/" indicates the sample lake boundary was only partly surveyed using the handheld GPS device.

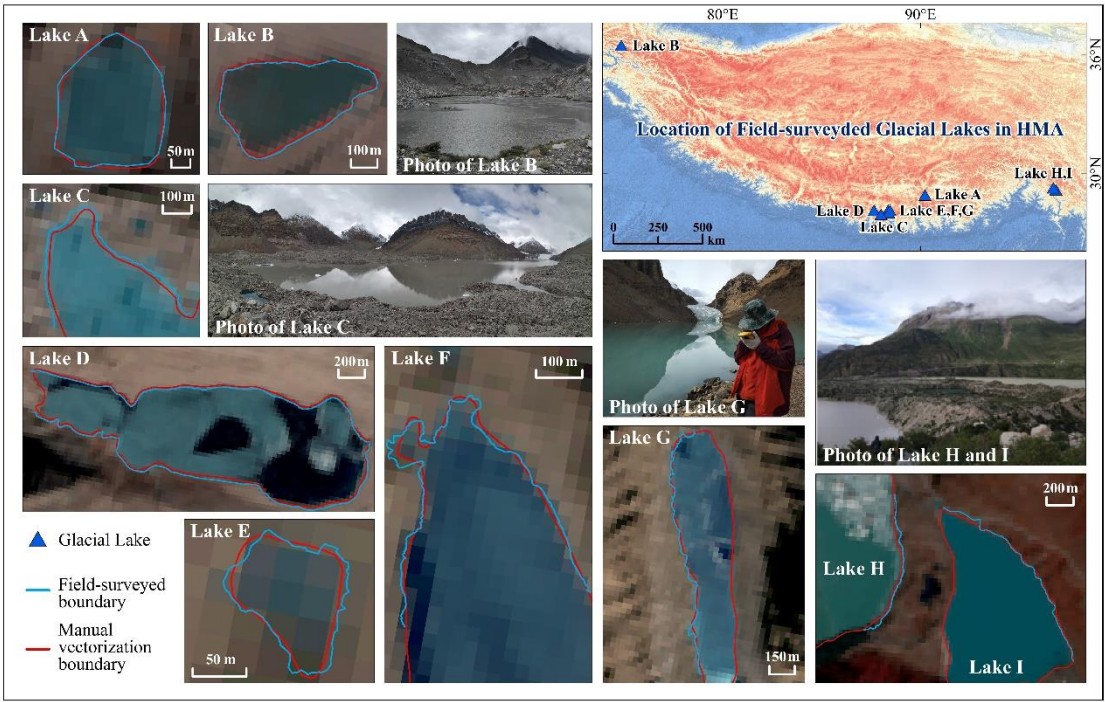

Figure 6. Glacial lakes in the HMA region surveyed in summer 2018 (the backgrounds of the surveyed lakes are Landsat OLI images).

## 6 Distribution and changes of HMA glacial lakes

As indicated by the achieved HMA glacial lake inventory, 30,121 (2080.12 ± 2.28 km$^2$) glacial lakes were identified in 2018 and their distribution had considerable spatial heterogeneity (Fig. 7). The greatest concentration of glacial lakes was in Altay and Sayan (335.42 ± 0.88 km$^2$, which accounted for 16.1 % of the total area of glacial lakes) and Eastern Himalaya (310.37 ± 0.89km$^2$, which accounted for 14.9 % of the total area of glacial lakes). Relatively few glacial lakes were found distributed in Eastern Kun Lun and Qilian Shan (38.85 ± 0.29 km$^2$, which accounted for 1.9 % of the total area of glacial lakes) and Eastern Tien Shan (40.55 ± 0.32 km$^2$, which accounted for 2.0 % of the total area of glacial lakes). The HMA glacial lakes were located within the elevation range of 1357–6247 m in 2018. An approximate normal distribution was presented both for the lakes of the entire HMA region and for the lakes in most sub-regions. More than 43 % of the HMA lake area has survived within the vertical range of 4500–5400 m, with the peak lake area of 241.89 ± 0.80 km$^2$ (accounting for 11.6 % of the total area) in

the range of 5100–5300 m. The elevation band of peak lake area in the different sub-regions varied from
2300–2500 m in Altay and Sayan to 5300–5500 m in Central Himalaya, Karakoram, and Western Kun
Lun.

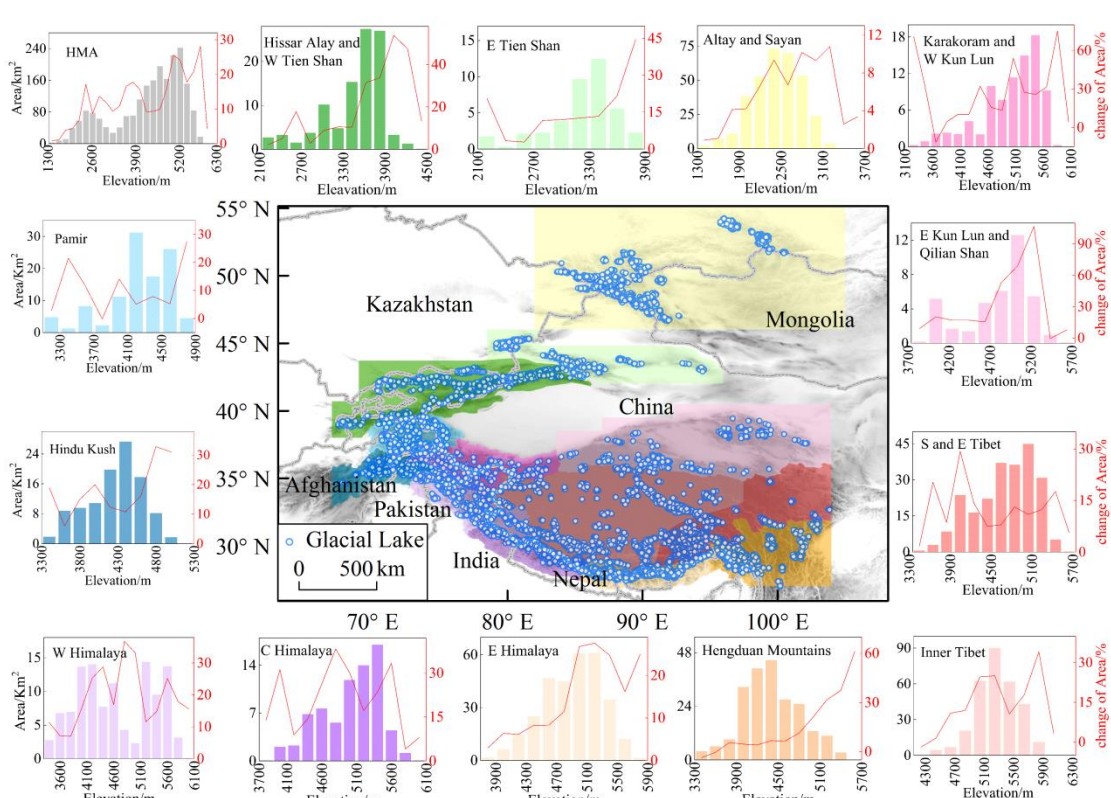

Figure 7. Distribution and change of glacial lake area in the entire HMA and its 13 sub-regions from

7              1990–2018.

The HMA glacial lakes experienced widespread areal expansion during 1990–2018 with an
average rate of increase in area of 15.2 % (Fig. 7). The rate of change of area varied widely between
different sub-regions and different 200 m elevation bands. The glacial lakes in Eastern Kun Lun and
Qilian Shan experienced the most rapid expansion in area during 1990–2018 with an average rate of
increase of 45.6 %, whereas the rate of change was only7.5 % in Altay and Sayan. Glacial lakes have
tended to develop to higher elevations during recent decades with the maximum distribution elevation
of 6078 m in 1990 rising to 6247 m in 2018. The rate of change of glacial lakes in the different 200 m
elevation bands presented a large average trend against elevation, rising as a whole during 1990–2018
(Fig. 7). The lake area expanded most from the elevation of approximately 5000 m and the rate of
expansion reached approximately 28 % at 5700–5900 m in the entire HMA region, although it differed
between different sub-regions. Lake area showed a notable rate of increase with elevation in most
sub-regions, e.g., Hissar Alay and Western Tien Shan, Hindu Kush, Eastern Himalaya, Hengduan
Mountains, Eastern Tien Shan, and Altay and Sayan. The rate of expansion varied markedly and no
observable trends in the rate of increase or decrease with elevation were discovered in Karakoram and
Western Kun Lun, Western Himalaya, and Inner Tibet. The rate of expansion in Central Himalaya and
Southern and Eastern Tibet was found to have seemingly decreased with increasing elevation (Fig. 7).
**7 Comparison and limitations**
There are at least 34 published reports or data sets on the regional extent of glacial lakes in the
HMA area, which are based on various lake boundary extraction methods and different data sources
(Supplementary Table S1). This previous research work examined glacial lakes from as early as 1962
up until 2017. However, it is difficult to evaluate any discrepancies comprehensively because different
extents of glacial lake distribution were examined and inconsistent thresholds of minimum lake area
were used. Glacial lake inventory data of the Third Pole region in 1990 (Zhang et al., 2015) and of the
HMA (Chen et al., 2020) in 2017 were used for comparison because both recorded glacial lakes in the
same buffer zone (i.e., within 10 km of the modern glacier extent) and over similar periods. For the
comparison, the same thresholds and regions were adopted for the inventory data. Marked
discrepancies were found to exist between the different datasets in terms of both the number and the
area of the glacial lakes. In 1990, only 4601 glacial lakes ($\geqslant$0.0054km$^2$) with total area of 554.33km$^2$
were recorded by Zhang et al. (2015), whereas 20,410 glacial lakes with total area of 1376.23 km$^2$ were
catalogued in the Third Pole region in this study. In 2017, 14,477 glacial lakes with total area of
1635.94 km$^2$ were recorded by Chen et al. (2020), whereas, we recorded 22,727 glacial lakes ($\geqslant$0.0081
km$^2$) with total area of 1726.41 km$^2$ in 2018 in HMA (excluding Altai and Sayan). We consider the
discrepancies attributable to three primary factors. (1) The buffer zone within 10 km of the modern
glacier extent was inconsistent between the data sets because different glacier inventories were used. (2)
Different operatives catalogued the glacial lakes using different remote sensing data covering different
periods. (3) Many glacial lakes were possibly missed because of the comparatively less manual

vectorization effort involved in the work of Zhang et al. (2015) and Chen et al. (2020). Overall, our glacial lake inventory catalogued glacial lakes throughout the entire HMA more comprehensively and with more careful error assessment when compared with available glacial lake data sets from regional or river-basin-based studies.

Several limitations deserve proper consideration when using the glacial lake inventory data. First, a degree of uncertainty resulted from using Landsat image data that covered different periods, i.e., both interannually and seasonally. Although images acquired in summer or autumn (June–November) were set as optimal choices, the selected images covered most seasons of the year, e.g., the images selected in June–November accounted for only 72.3 and 88.8 % of the total number in 2018 and 1990, respectively. Interannually, images were selected from a span of 10 years (1986–1995) and 4 years (2016–2019) to obtain sufficient high-quality images of the HMA area. Second, this study recorded all lakes located within the 10 km buffer area of glacier extent as glacial lakes. Therefore, certain lakes that have no relation to glaciers or to glaciation (i.e., non-glacial lakes) in the alpine cryosphere were potentially catalogued in error because of the difficulty in distinguishing non-glacial lakes from glacial lakes based on remote sensing data. Third, we identified water bodies related to glaciers or to glaciation in the alpine cryosphere as glacial lakes. However, in many cases, it was difficult to determine whether such bodies should be recorded as glacial lakes, e.g., cases of long narrow water bodies on rivers and cases where the number of pure water body pixels was small. Thus, some errors and inconsistences were inevitable because of having different operatives performing the lake boundary vectorization and inspection. In future, this glacial lake inventory will be updated and shared on the National Special Environment and Function of Observation and Research Stations Shared Service Platform (China).

**8 Data availability**

The data set developed in this study comprises two .shp file documents containing the glacial lake inventory of the HMA region in 1990 and 2018. The data set can now be accessed via the website of the National Special Environment and Function of Observation and Research Stations Shared Service Platform (China): http://www.crensed.ac.cn/portal/metadata/706ce17f-1684-4e8d-bf5e-7d517e03693c (Wang et al., 2019a).

**9 Conclusions**

A glacial lake inventory of the HMA region was realized based on satellite remote sensing data and GIS techniques. Eight attribute items were recorded in the glacial lake inventory data set of the

HMA region. Lake area error was assessed carefully with respect to theoretical analysis of lake boundary pixels and actual boundaries derived by GPS field-based surveys. On average, the deviations between the delineation of lake boundaries derived using the two methods were within ±0.5 pixels (±15 m). The relative area errors of each lake in 2018 varied from 1–79 %, and the average relative area errors of ±13.2 % in the entire HMA region were characterized by increase in the relative area error with decreasing lake size.

Overall, 30,121 glacial lakes with a total area of 2080.12 ± 2.28 km$^2$ were catalogued in 2018 in the HMA region. Glacial lakes survived in all 13 sub-regions of HMA from the elevation of 1357 to 6247 m. Glacial lakes were found concentrated in the sub-regions of Altay and Sayan and Eastern Himalaya and at elevation bands of 4500–5400 m. The HMA glacial lakes have experienced widespread expansion with an average rate of increase in area of 15.2%. Lake area expanded most in the higher elevation bands during 1990–2018. The data set is expected to provide basic data to support cryosphere hydrology research, water resources utilization and management, and assessment of glacier-related hazards in the HMA region.

## 10 Acknowledgements

The study was funded by the National Natural Science Foundation of China (No. 41771075, No. 41571061, and No. 41271091). The authors are grateful to the lake boundary vectorization operators who were not included as authors: Yao Chao, Chen Shiyin, Zhu Xiaoxi, Li Ruijia, Huang Rong, Peng Xin, Xiang Lili, Yi Ying, Liu Yanlin, Fu Yongqiao, Ran Weijie, and Gu Ju. We thank James Buxton MSc from Liwen Bianji, Edanz Group China (https://en-author-services.edanzgroup.com/), for editing the English text of this manuscript.

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
