# Peer review of "Glacial lake inventory of High Mountain Asia in 1990 and 2018 derived from Landsat images"

_Earth System Science Data, 2019_

## Referee Comment (RC1) · Anonymous Referee #1 · 1 Mar 2020

The present study provides an important glacial lake dataset across the High Mountain Asia using the Landsat images and manual visual interpretation. It will be served as a baseline for the further studies, such as changes in regional water resources and glacier-related hazard, especially the GLOFs, etc. The paper was well organized and written but there are still a lot of issues that need to be resolved before publishing in ESSD.

General comments:

1.The authors used the glacier boundaries and region divisions from RGI dataset 5.0. We know that both have been updated to the 6th version in 2017. Despite the small differences between the two versions in glacier divisions, the authors should carefully examine the differences in glacier boundaries between the two versions and use the

latest version as a basis because it is important for an up-to-date glacial lake inventory. Also, the authors merged some subregions in this study, which is different with the original region divisions in RGI dataset 5.0. I suggest the authors use the region divisions provided by RGI dataset 6.0 directly, if not, the authors should give the reasons for such mergers.

2.The authors got the total uncertainty of the lake area for the entire study region or subregions only by adding the uncertainty of each lake area. It would be wrong because the accumulation of errors should be based on error propagation theory rather than simple addition. I suggest the authors to download the document from the link: http://ipl.physics.harvard.edu/wp-uploads/2013/03/PS3_Error_Propagation_sp13.pdf, which include the detailed introductions on how errors are propagated.

3.For the High Mountain Asia, there are a lot of regional or river basin-based studies have been made on dynamics and evolutions of glacial lakes, and their potential hazard and risk assessments also primarily based on satellite images and GIS technology. Based on this, many glacial lake datasets have been produced, hence I suggest the authors add a sub-section at the end to collect and compare these regional or basin-based datasets with the dataset produced by the authors. It is important for data paper and will improve this manuscript.

4.The language of this paper still needs further polishing due to some inappropriate sentence's constructions. I had trouble in understanding and following some sentences, and suggest seeking a professional editor before publication.

Specific comments:

P1 L1: Please rephrase the title so that it can contain a more explicit time information because the present title seems to be a long time series dataset, but the authors only provided two periods of dataset.

P3 L28: . . . are shown in Figure 1.

P3 L29: It would be better to include the specific latitude and longitude ranges.

P4 L7: Add the reference(s) for the annual average glacier meltwater.

P6 L15: 4.2

P7 L5: Landsat TM/ETM+.

P7 L11-13: Provide more information about this step, i.e., how to determine the thresholds for different regions?

P7 L14: For the High Mountain Asia, a lot of regional or basin-based studies on glacial lakes have been made during the past decades. Hence, it is a better choice that the authors can collect these published glacial lake data to help identify and locate the glacial lakes except for the method the authors used because we know the water bodies automatically extracted contains many errors because of mountain shadows and snow cover.

P9 L3: Please check this reference format: Weicai et al., 2014

P10 L25-27: Please provide more details on how to use these ancillary data to distinguish the glacial lake types because it is important as a guide for similar studies in the future and in fact, we know that the subsurface channels are ubiquitous. Have the authors considered this problem and how to solve it?

P16 L3-4: This description is the opposite of that in the legend in Figure 6. In fact, it can be removed because it is already clear in the legend.

---

## Referee Comment (RC2) · Anonymous Referee #2 · 30 May 2020

This manuscript firstly introduces glacial lake inventory across the entire High Mountain Asia at two time periods using manual mapping on 30 m Landsat images. The sharing data are crucial to further water resource assessment or glacier hazard risk, even current lake data are not perfect, however, this is a big step toward data sharing at such a large-scale. I am sure more and better glacial lake data will be shared or updated by this team or other research group in the future, inspired by this original report. Before recommending this manuscript to be published in ESSD, I would like to suggest some necessary revision.

General comments:

1. Currently, the authors give a wrong link to download the sharing data that is unavailable or need a registered account to download the data, it is

not convenient. Actually, I find it is okay to directly download without any registration at http://www.crensed.ac.cn/portal/metadata/706ce17f-1684-4e8d-bf5e-7d517e03693c, so, why not replace with this link in the text?

2. There are many reference errors in the current text that have to be corrected carefully, I did not find all of them out, the authors have to go through. Such as, Yang et al., 2019 in the text but not listed in the reference section, P19-L24, Chaohai L, wrong surname, the same to P20-L1, L4, P21-L20, P23-L10.

3. A technical question, how do you distinguish non-glacial lakes within 10-km of reference glaciers with defined glacial lake? For example, the 2010 Ataabad landslide-dammed lake. In the alpine area of HMA, there are many landslide-dammed lakes that have no relationship with glaciology but are mainly supplied by glacier meltwater. Could you provide more detailed information about this?

4. About the minimum mapping unit, how do you consider selecting 0.0054 km2 as a threshold value? Currently written as "the minimum glacial lake area recorded was set at 0.0054 km2 (e.g., 3–4 pure 10 lake water body pixels with approximately 12 mixed boundary pixels) because a lake area covering 11 fewer than three pure lake water pixels could possibly have an error of >100 %", I am confused by this writing. My understanding is that it is difficult to digitize 3-4 pixels by manual interpretation. Otherwise, it is 6 pixels, equaling to 0.0054 km2 using Landsat images.

5. I suggest writing a further revision plan in the end of this draft to point out the shortage of current glacial lake data. Actually, the data have been published in a data sharing platform, and some errors exist inevitably in terms of two times manually vectorization for the same lake, for example, induced by wrongly digitizing, maybe operated by different operatives. Do you have any plan to update the data? Once the data updated where to share? In what kind of ways?

6. I also suggest authors to polish the language once again, and some sentences are arduous to follow.

Specific comments: P1-L26, update the link

P2-L25, recognizing→revealing

P3-L6, Yang et al., 2019, not listing in the reference; L18, a Landsat imagery series?

P4-L5, only Antarctic? What about Arctic? L8 "the primary source of both lake basin formation", I think no relationship.

P5-L13, suggest revise as circa 1990 and circa 2018

P6-L19, 20, what scale do you keep while an operator did computer screen vectorization of mixed pixels?

P7-L12, -0.1 of NDWI to extract lake extent? Generally, this value is greater than 0.

P9-L3, Weicai et al.,?

P10-L3, 10 km from modern glacier terminals? Or glacier extent? L12-15, given a reasonable classification of lakes, why did not you take this? L19-22, it is not clear what your point is? "because of the lack of sufficient amounts of remote sensing data with appropriate resolution." L23, two types: glacier-fed lakes and non-glacier-fed lakes? The significance becomes very limited by too simple classification system. Maybe more types, such as pro-connected lake and supraglacial lake, be cataloged. But being cautious, once you modified the data, meaning that you have to update the sharing data on the platform online.

P13-L14,15, why did not you record the date of used images? Only recorded the month and year?

P14-L21, Narrate the accuracy of Trimble GeoXH6000 for a better understanding about the validation.

P16, Figure 6, suggest adding a scale bar for each subset.

P16-L13 The HMA glacial lakes are located within the elevation range of 1600–6300

m. while, P17-L110, L11, maximum distribution elevation of 6078 m in 1990 rising to 6247 m in 2018. Maybe use the relatively accurate value of elevation.

P17, Back up to previous error, in Figure 7, the maximum X axis value is 6000 m, so you miss your lakes with maximum elelevation.

P18, L1, L2, How to prove that no observable trends were discovered in Karakoram and Western Kun Lun, Western Himalaya?

P18, suggest adding a section about the shortage and updating plan for this data, putting before Data availability

P18, replacing the existing link

P18, rewrite the sentences in L23-26, it is unclear. "Lake area expanded most in the higher elevation bands during 1990–2018. The data set has been developed as basic data for cryosphere hydrology research; however, it is expected that it could support practical utilization and management of water resources and assessment of glacier-related hazards in the HMA region"

---

## Author Comment (AC1) · 8 Jul 2020

**General comments:**

1.The authors used the glacier boundaries and region divisions from RGI dataset 5.0. We know that both have been updated to the 6th version in 2017. Despite the small differences between the two versions in glacier divisions, the authors should carefully examine the differences in glacier boundaries between the two versions and use the latest version as a basis because it is important for an up-to-date glacial lake inventory. Also, the authors merged some subregions in this study, which is different with the original region divisions in RGI dataset 5.0. I suggest the authors use the region divisions provided by RGI dataset 6.0 directly, if not, the authors should give the reasons for such mergers.

Based on your suggestion, the RGI dataset 6.0 has been used to recalculate the buffer zone of 10 km from modern glacier terminals in the revised version. We have carefully reexamined and updated the glacial lake inventory of HMA based on the buffer zone of 10 km from modern glacier terminals of RGI dataset 6.0. All the descriptions and results based on RGI 5.0 in the previous version have been updated simultaneously. In all, the total area of buffer zone increased from 1.19 × $10^6$ km$^2$ to 1.25 × $10^6$ km$^2$, and new Landsat images have been collected to fill the gaps between the two buffer zones calculated by RGI dataset 5.0 and 6.0 respectively. 1117 and 1169 glacier lakes with a total area of 113.77 km$^2$ and of 124.22 km$^2$ in 1990 and 2018 have been newly recorded in a 10 km buffer area of glacier terminals of RGI 6.0 respectively compared with in a 10 km buffer area of glacier terminals of RGI 5.0. The updated data is shared at http://www.crensed.ac.cn/portal/metadata/706ce17f-1684-4e8d-bf5e-7d517e03693c.

There are 16 subregions in HMA region according to RGI dataset 6.0. However, relatively fewer glacier lakes in 2018 survived in some subregions such as 235 and 572 glacial lakes with an area of 13.94 and of 24.91 km$^2$ in Qilian Shan and E Kun Lun, 264 and 706 lakes with an area of 44.97 and of 47.29 km$^2$ in W Kun Lun Shan and Karakoram, 200 and 1624 lakes with an area of 11.01 and of 89.99 km$^2$ in Hissar Alay and W Tien Shan. Characteristics presented by too small sample lakes in these subregions appeared incongruous. Thus, E Kun Lun and Qilian Shan, Hissar Alay and W Tien Shan, Karakoram and W Kun Lun were merged respectively according to geomorphology and their climatic background characteristics. The rest of 10 subregions are the same as RGI dataset 6.0. Finally, we get 13 subregions in this study.

2.The authors got the total uncertainty of the lake area for the entire study region or subregions only by adding the uncertainty of each lake area. It would be wrong because the accumulation of errors should be based on error propagation theory rather than simple addition. I suggest the authors to download the document from the link:http://ipl.physics.harvard.edu/wpuploads/2013/03/PS3_Error_Propagation_sp13.pdf, which include the detailed introductions on how errors are propagated.

Yes, error propagation wasn't considered in the previous version. A single lake area uncertainty may be either overestimated or underestimated by the formula (2) of the manuscript. Thus, it was wrong to calculate the total uncertainty only by adding the uncertainty of each lake area. We have recalculated uncertainty of the lake area for the entire study region or subregions by the following formula according to the suggested document (link:http://ipl.physics.harvard.edu/wpuploads/2013/03/PS3_Error_Propagation_sp13.pdf):

$$E_T = \sqrt{\sum_{i=1}^{n} a_i{}^2}$$

where "$E_T$" is the area error of the entire study region or subregions, "i" is number of the lake in the entire study region or subregions, and "a" is the error area of a single lake. We have updated the uncertainty contexts in line of P15 L1–5 in the revised manuscript.

3. For the High Mountain Asia, there are a lot of regional or river basin-based studies have been made on dynamics and evolutions of glacial lakes, and their potential hazard and risk assessments also primarily based on satellite images and GIS technology. Based on this, many glacial lake datasets have been produced, hence I suggest the authors add a sub-section at the end to collect and compare these regional or basin-based datasets with the dataset produced by the authors. It is important for data paper and will improve this manuscript.

This is a very precious suggestion. We have accordingly collected the available documents or datasets investigating the glacial lake in HMA and have excerpted a Supplementary Table S1. The description about dataset comparisons has been added in a sub-section of "**comparison and limitation**"(P19 L11-P20 L21).

There are at least 34 published reports or datasets on the regional extent of glacial lakes in the HMA area, which are based on various lake boundary extraction methods and different data sources (see Supplementary Table S1). The previous research work examined glacial lakes from as early as 1962 up until 2017. However, it is difficult to evaluate any discrepancy comprehensively because glacial lake distribution was examined in different extents and thresholds of minimum lake area were used inconsistently. Therefore, glacial lake inventory data of the Third Pole region in 1990 (Zhang et al., 2015) and of the HMA (Chen et al., 2020) in 2017 have been used for comparison because both recorded glacial lakes in the same buffer zone (i.e., within 10 km of the modern glacier extent) and over similar periods. For the comparison, same thresholds and regions have been adopted for the inventory data. Marked discrepancies have been found to exist between the different datasets in terms of both the number and the area of the glacial lakes. In 1990, only 4601 glacial lakes ($\geqslant$0.0054km$^2$) with a total area of 554.33km$^2$ were recorded by Zhang et al. (2015), whereas 20,410 glacial lakes with a total area of 1376.23 km$^2$ have been catalogued in the Third Pole region in this study. In 2017, 14,477 glacial lakes with a total area of 1635.94 km$^2$ were recorded by Chen et al. (2020), whereas, we have recorded 22,727 glacial lakes ($\geqslant$0.0081 km$^2$) with a total area of 1726.41 km$^2$ in 2018 in HMA (excluding Altai and Sayan). We consider the discrepancies attributable to three primary factors. (1) The buffer zone within 10 km of the modern glacier extent is inconsistent between the data sets because different glacier inventories have been used. (2) Different operatives have catalogued the glacial lakes using different remote sensing data covering different periods. (3) Many glacial lakes were possibly missed because of comparatively less manual vectorization effort involved in the work of Zhang et al. (2015) and Chen et al. (2020). Overall, our glacial lake inventory has catalogued glacial lakes throughout the entire HMA more comprehensively and with more careful error assessment compared with available glacial lake data sets from regional or river-basin-based studies.

Table 1. The comparison of glacial lake amount from the documents of Zhang et al. (2015) and Chen et al. (2020) with that from this manuscript

| Region | Year | Numbers | Area (km²) | Minimum Area (km²) | Reference |
|---|---|---|---|---|---|
| The Third Pole region | 1990 | 4601 | 554.33 | 0.0054 | Zhang et al., 2015 |
| | 1990 | 20410 | 1376.23 | | Wang et al., 2020 |
| HMA (Altai mountains excluded) | 2017 | 14477 | 1635.94 | 0.0081 | Chen et al., 2020 |
| | 2018 | 22727 | 1726.41 | | Wang et al., 2020 |

4.The language of this paper still needs further polishing due to some inappropriate sentence's constructions. I had trouble in understanding and following some sentences, and suggest seeking a professional editor before publication.

The language of the revised manuscript has been polished by a professional language retouching company.

**Specific comments:**

P1 L1: Please rephrase the title so that it can contain a more explicit time information because the present title seems to be a long time series dataset, but the authors only provided two periods of dataset.

The title of manuscript has been changed into "Glacial lake inventory of High Mountain Asia in 1990 and 2018 derived from Landsat images"

P3 L28: : : : are shown in Figure 1.

It has been revised.

P3 L29: It would be better to include the specific latitude and longitude ranges.

It has been revised as "This region (26°–54°N, 67°–104°E)"

P4 L7: Add the reference(s) for the annual average glacier meltwater.

We are sorry to say that we cannot find the source of "an average meltwater volume of 110–150 km³ a⁻¹". For this reason, we have used a new data of glacier negative mass balance of -150±110 kg m⁻² a⁻¹ in HAM to replace it. This information is sourced from "Hock, R., Rasul G., Adler C., Cáceres B., Gruber S., Hirabayashi Y., Jackson M., Kääb A., Kang S., Kutuzov S., Milner A., Molau U., Morin S., Orlove B., and Steltzer H. 2019. High Mountain Areas. In: IPCC Special Report on the Ocean and Cryosphere in a Changing Climate [Pörtner H.-O., Roberts D.C., Masson-Delmotte V., Zhai P., Tignor M., Poloczanska E., Mintenbeck K., Alegría A., Nicolai M., Okem A., Petzold J., Rama B., Weyer N.M. (eds.)]".

P6 L15: 4.2

It has been revised.

P7 L5: Landsat TM/ETM+.

It has been revised.

P7 L11-13: Provide more information about this step, i.e., how to determine the thresholds for

different regions?

In the present study, we have determined the optimal thresholds in each region or image. By considering the edge effects according to the mixed pixels, this study firstly selects a lower optimal threshold (approx. −0.1) for specific images to obtain the maximum water body. Then, higher thresholds are tested for visual water extraction before a suitable threshold (varied in the range of −0.10 to 0.20) is selected. It has been explained in line P7 L12–19.

P7 L14: For the High Mountain Asia, a lot of regional or basin-based studies on glacial lakes have been made during the past decades. Hence, it is a better choice that the authors can collect these published glacial lake data to help identify and locate the glacial lakes except for the method the authors used because we know the water bodies automatically extracted contains many errors because of mountain shadows and snow cover.

This is a valuable suggestion. We have accordingly collected and used the available glacial lake data in HMA (see Supplementary Table S1) to identify and locate the glacial lakes when the glacial lake inventory of HMA was reexamined and updated.

P9 L3: Please check this reference format: Weicai et al., 2014

It has been revised as "Wang et al., 2014".

P10 L25-27: Please provide more details on how to use these ancillary data to distinguish the glacial lake types because it is important as a guide for similar studies in the future and in fact, we know that the subsurface channels are ubiquitous. Have the authors considered this problem and how to solve it?

We have distinguished the glacial lake types of glacier-fed from the non-glacier-fed by whether or not a glacial lake can possibly receive surface meltwater from the modern glacier (Fig. 1). We have recorded a glacier-fed lake based on the following facts: (1) a lake has a lower elevation than modern glacier (mother glacier); (2) the mother glacier(s) melting water can visually flow into lake through surface flow route assisted by 3D digital terrain imagery from Google Earth; (3) all the glacial lakes were visually examined one by one.

Theoretically, the boundary of glacial lake basin and melting water surface flow route can be calculated based on DEM data which undoubtedly would contribute to distinguishing the glacier-fed lake from the non-glacier-fed lake. Practically, we tried but failed to do this at the present stage for no appropriate DEM data with satisfactory resolution was obtained since so many small glacial lakes survived in HMA. Nevertheless, we will further focus on this in the future work.

The glacial lake fed by melting water through subsurface channels is a common phenomenon and little field-surveyed work about this has been reported. We choose to ignore this issue as it is difficult to survey the subsurface channels of glacial lakes from remote sensing data. In addition, lake type is distinguished from topographic features of the lake basin and modern glaciers. In most cases, the lake can be possibly fed by melting water both from subsurface channels and surface route.

We have added the explanation in line P11 L1–9.

[Figure]

[Figure]

Fig. 1 Different types of glacial lakes distinguished from 3D digital terrain imagery from Google Earth

P16 L3-4: This description is the opposite of that in the legend in Figure 6. In fact, it can be removed because it is already clear in the legend.

It has been deleted.

---

## Author Comment (AC2) · 8 Jul 2020

Thank you very much for your valuable suggestions and your hard work since the submission of our paper. We have responded to the comments of both Anonymous Referee. Our responses were attached separately by supplement files in red font. Three supplement files were submitted, i.e., responses to Anonymous Referee #1, responses to Anonymous Referee #2, and a Supplementary Table S1.

Please also note the supplement to this comment:
https://essd.copernicus.org/preprints/essd-2019-212/essd-2019-212-AC2-supplement.zip

---

## Author Comment (AC3) · 8 Jul 2020

**General comments:**

1. Currently, the authors give a wrong link to download the sharing data that is unavailable or need a registered account to download the data, it is not convenient. Actually, I find it is okay to directly download without any registration at http://www.crensed.ac.cn/portal/metadata/706ce17f-1684-4e8d-bf5e-7d517e03693c, so, why not replace with this link in the text?

    It has been replaced by the new link.

2. There are many reference errors in the current text that have to be corrected carefully, I did not find all of them out, the authors have to go through. Such as, Yang et al., 2019 in the text but not listed in the reference section, P19-L24, Chaohai L, wrong surname, the same to P20-L1, L4, P21-L20, P23-L10.

    We have carefully rechecked all the references in the text and reference list and corrected the errors and mistakes.

3. A technical question, how do you distinguish non-glacial lakes within 10-km of reference glaciers with defined glacial lake? For example, the 2010 Ataabad landslide dammed lake. In the alpine area of HMA, there are many landslide-dammed lakes that have no relationship with glaciology but are mainly supplied by glacier meltwater. Could you provide more detailed information about this?

    The lakes related to glaciers or glaciation in the alpine cryosphere (defined within 10 km-buffer area of glacier terminals) have all been recorded as glacial lakes. Operationally, all the lakes located at the 10 km-buffer area of glacier extents have been recorded as glacial lakes in this manuscript. Some lakes which have no relation to glaciers or glaciation (i.e., non-glacial lakes) in the alpine cryosphere were possibly cataloged. We arbitrarily think the number of non-glacial lakes is relatively small and recorded all the lakes within 10 km-buffer area of glacier extents as (1) it is difficult to distinguish these non-glacial lakes from glacial lakes based on remote sensing data and (2) recoding all the lakes is conducive to evaluating the entire water source in the alpine cryosphere. Thus, the non-glacial lakes are usually not distinguished if located near the modern glaciers when glacial lake inventory is carried out in most documents.

    We have tried to exclude the lake formed by man-made dam and/or long and narrow water body on river which could be river flood or wetland distributed along river. The Ataabad lake dammed by landslide in Huza river has been fed by glacier melting water since it was formed in Jan., 2010. By definition, the naturally formed Ataabad lake is located within 10 km-buffer area of glacier extent and received melting water should be recorded as glacial lake. Unfortunately, Ataabad lake was falsely catalogued in the previous inventory due to its long and narrow shape (~23km long). We have added the Ataabad lake in our updated glacial lake inventory of HMA. Furthermore, the long and narrow water bodies (usually from several kilometers long to tens) developed along river have been reexamined in the updated glacial lake inventory.

    The technical comments have been discussed in the adding section of "7 comparison and limitation".

4. About the minimum mapping unit, how do you consider selecting $0.0054$ km$^2$ as a threshold value? Currently written as "the minimum glacial lake area recorded was set at $0.0054$ km$^2$ (e.g., 3–4 pure lake water body pixels with approximately 12 mixed boundary pixels) because a lake area covering fewer than three pure lake water pixels could possibly have an error of >100 %", I am confused by this writing. My understanding is that it is difficult to digitize 3-4 pixels by manual

interpretation. Otherwise, it is 6 pixels, equaling to 0.0054 km$^2$ using Landsat images.

I am sorry to make mistakes about the expressions. The manual delineation was required for approximately 1/2, 1/8, or 7/8 of the peripheral mixed pixels surrounding pure lake water body pixels. For example, in Fig. 1A, three pure lake water body pixels were possibly surrounded by maximum twelve peripheral mixed pixels. Theoretically, the ratio of pure lake water body pixels area to mixed peripheral mixed pixels area can be:

$$\frac{\text{Area of peripheral mixed pixels}}{\text{Area of three pure lake water body pixels}} \times 100\% = \frac{6 \times \frac{1}{2} + 5 \times \frac{1}{8} + 1 \times \frac{7}{8}}{3} \times 100\% = 150\%.$$

Similarly, in Fig. 1B, four pure lake water body pixels were also possibly surrounded by maximum twelve peripheral mixed pixels, and the ratio of pure lake water body pixels area to mixed peripheral mixed pixels area can be:

$$\frac{\text{Area of peripheral mixed pixels}}{\text{Area of four pure lake water body pixels}} \times 100\% = \frac{8 \times \frac{1}{2} + 4 \times \frac{1}{8}}{4} \times 100\% = 112.5\%.$$

Then when area error within one standard deviation (1σ) (Hanshaw and Bookhagen, 2014) was considered (i.e., the error adjusted coefficient of 0.6872 was used), the error (1σ) of lake area with three pure lake water body pixels and four lake water body pixels can be expressed as:

$$E = \frac{\text{Area of peripheral mixed pixels} \times 0.6872}{\text{Area of three pure lake water body pixels}} \times 100\% = 1.5 \times 0.6872 \times 100\%$$
$$= 103.1\%.$$

And

$$E = \frac{\text{Area of peripheral mixed pixels} \times 0.6872}{\text{Area of four pure lake water body pixels}} \times 100\% = 1.125 \times 0.6872 \times 100\%$$
$$= 77.3\%.$$

So, in the Landsat images, 3–4 pure lake water pixels could possibly be surrounded by approximately 12 mixed boundary pixels with the total area equivalent to about 3 peripheral mixed pixels, i.e., 6 pixels area (equals to 0.0054 km$^2$) or more in total lake water body. When the threshold of 3–4 pure lake water pixels is set as minimum recorded lake water body, the uncertainty could theoretically amount to about 100%, and a lake area covering fewer than three pure lake water pixels could possibly have a relative error of >100 %.

We have further rewritten the arduous expressions (P12–L16–30) and Figure 4. has also been reedited. (P13–L1–8)

[Figure]

Fig. 1 Sketch maps showing the 3–4 pure lake water body pixels with approximately 12 mixed boundary pixels

Reference:

Hanshaw, M. N., and Bookhagen, B.: Glacial areas, lake areas, and snow lines from 1975 to 2012: status of the Cordillera Vilcanota, including the Quelccaya Ice Cap, northern central Andes, Peru, The Cryosphere, 8, 359-376, doi:10.5194/tc-8-359-2014, 2014.

5.I suggest writing a further revision plan in the end of this draft to point out the shortage of current glacial lake data. Actually, the data have been published in a data sharing platform, and some errors exist inevitably in terms of two times manually vectorization for the same lake, for example, induced by wrongly digitizing, maybe operated by different operatives. Do you have any plan to update the data? Once the data updated where to share? In what kind of ways?

Several limitations deserve proper consideration when using the glacial lake inventory data. First, a degree of uncertainty has resulted from using Landsat image data that covered different periods, i.e., both interannually and seasonally. Although images acquired in summer or autumn (June–November) have been set as optimal choices, the selected images covered most seasons of the year, e.g., the images selected in June–November accounted for only 72.3 and 88.8 % of the total number in 2018 and 1990, respectively. Interannually, images were selected from a span of 10 years (1986–1995) and 4 years (2016–2019) to obtain sufficient high-quality images of the HMA area. Second, this study has recorded all lakes located within the 10 km buffer area of glacier extent as glacial lakes. Therefore, certain lakes that have no relation to glaciers or glaciation (i.e., non-glacial lakes) in the alpine cryosphere were potentially catalogued in error because of the difficulty in distinguishing non-glacial lakes from glacial lakes based on remote sensing data. Third, we have identified water bodies related to glaciers or glaciation in the alpine cryosphere as glacial lakes. However, in many cases, it was difficult to determine whether such bodies should be recorded as glacial lakes, e.g., cases of long narrow water bodies on rivers and cases where the number of pure water body pixels was small. Thus, some errors and inconsistences were inevitable because the lake boundary vectorization and inspection were performed by different operatives. In the future, this glacial lake inventory will be updated and shared on the National Special Environment and Function of Observation and Research Stations Shared Service Platform (China), and further water source evaluation and hazards assessment would be carried out in our next research schedule.

The main limitations of current glacial lake inventory data have been added in line (P20–L5–21).

6. I also suggest authors to polish the language once again, and some sentences are arduous to follow.

The language of the revised manuscript has been polished by a professional language retouching company.

**Specific comments:**

P1-L26, update the link

It has been updated.

P2-L25, recognizing→revealing

It has been modified.

P3-L6, Yang et al., 2019, not listing in the reference; L18, a Landsat imagery series?

The reference of Yang et al., 2019 has been added; "a Landsat imagery series" has been

modified as "Landsat images"

P4-L5, only Antarctic? What about Arctic? L8 "the primary source of both lake basin formation", I think no relationship.

L5: It has been modified as "The HMA area has the largest surviving glaciers of any region other than the polar regions"

L8: It has been modified as "which was the primary source of water supply for the development of glacial lakes"

P5-L13, suggest revise as circa 1990 and circa 2018.

It has been revised.

P6-L19, 20, what scale do you keep while an operator did computer screen vectorization of mixed pixels?

The viewing scale of 1:10000 has been kept and it has been modified as "e.g., viewing scale of 1:10,000 on a computer screen vectorization of mixed pixels".

P7-L12, -0.1 of NDWI to extract lake extent? Generally, this value is greater than 0.

Liu et al. (2016) and Du et al. (2014) suggested that it might be preferable to set the optimized threshold of $NDWI_{GREEN/NIR}$ of Landsat OLI images to $-0.05$. By considering the edge effects according to the mixed pixels, this study has initially selected a lower optimal threshold (approx. $-0.1$) for specific images to obtain the maximum water body (Fig. 2). Then, higher thresholds have been tested for visual water extraction before selecting the suitable threshold (varied in the range of $-0.10$ to $0.20$).

[Figure]

Fig. 2 An example showing the optimal thresholds for $NDWI_{GREEN/NIR}$. (a) original OLI image taken in 2018; (b) the $NDWI_{GREEN/NIR}$ image

Reference:

Du, Z., Li, W., Zhou, D., Tian, L., Ling, F., Wang, H., Gui, Y., and Sun, B.: Analysis of Landsat-8 OLI imagery for land surface water mapping, Remote Sens. Lett., 5, 672-681, doi:10.1080/2150704x.2014.960606, 2014.

Liu, Z., Yao, Z., and Wang, R.: Assessing methods of identifying open water bodies using Landsat 8 OLI imagery, Environ. Earth Sci., 75, doi:10.1007/s12665-016-5686-2, 2016.

P9-L3, Weicai et al.,?

It has been revised as "Wang et al., 2014".

P10-L3, 10 km from modern glacier terminals? Or glacier extent?

The "glacier extent" may be more exact and the relative expressions has been revised.

P10-L12-15, given a reasonable classification of lakes, why did not you take this?

The classification schema by Yao et al. (2018) is difficult to carry out based on available remote sensing data for so large area as HMA. For it is a little difficult to distinguish glacial lake type in terms of material properties, topographic features, and phase of lake formation using remote sensing imagery.

P10-L19-22, it is not clear what your point is? "because of the lack of sufficient amounts of remote sensing data with appropriate resolution."

It has been revised as "because of the lack of remote sensing data with satisfied spatial resolution."

P10-L23, two types: glacier-fed lakes and non-glacier-fed lakes? The significance becomes very limited by too simple classification system. Maybe more types, such as pro-connected lake and supraglacial lake, be cataloged. But being cautious, once you modified the data, meaning that you have to update the sharing data on the platform online.

The glacier-fed lakes were further subdivided into three sub-classes: supraglacial lakes (lakes developed on glacier surface), ice-contacted lakes (lakes contacting the glacier terminal or margin), and ice-uncontacted lakes (lakes not contacting the glacier but fed directly by glacial meltwater). We have updated attribute items in the datasets and resubmitted them on the platform online (http://www.crensed.ac.cn/portal/metadata/706ce17f-1684-4e8d-bf5e-7d517e03693c).

It has been added in P10–L24-27.

P13-L14,15, why did not you record the date of used images? Only recorded the month and year?

The date of used images has been input in the attribute item of Lake time phase.

P14-L21, Narrate the accuracy of Trimble GeoXH6000 for a better understanding about the validation.

The accuracy of Trimble GeoXH6000 is in decimeter and has been added in line of P15–L18, L19.

P16, Figure 6, suggest adding a scale bar for each subset.

It has been added.

P16-L13 The HMA glacial lakes are located within the elevation range of 1600–6300m. while, P17-L10, L11, maximum distribution elevation of 6078 m in 1990 rising to 6247 m in 2018. Maybe use the relatively accurate value of elevation.

The accurate elevation value of 1357–6247m has been used to replace the elevation range of 1600–6300m in P17–L13.

P17, Back up to previous error, in Figure 7, the maximum X axis value is 6000 m, so you miss your

lakes with maximum elevation.

I am sorry to miss the maximum X axis value and it has been updated to 6300m.

P18, L1, L2, How to prove that no observable trends were discovered in Karakoram and Western Kun Lun, Western Himalaya?

I am sorry that we presented an arduous expression. The sentence has been revised as "The expansion rate varied markedly and no observable trend in the rate of increase or decrease with elevation were discovered in Karakoram and Western Kun Lun, Western Himalaya, and Inner Tibet".

P18, suggest adding a section about the shortage and updating plan for this data, putting before Data availability

A section about the comparison, shortage and updating plan for this data has been added.

P18, replacing the existing link

It has been replaced.

P18, rewrite the sentences in L23-26, it is unclear. "Lake area expanded most in the higher elevation bands during 1990–2018. The data set has been developed as basic data for cryosphere hydrology research; however, it is expected that it could support practical utilization and management of water resources and assessment of glacier-related hazards in the HMA region"

It has been rewritten as "The data set is expected to provide basic data to support the cryosphere hydrology research, water resources utilization and management, and assessment of glacier-related hazards in the HMA region."

---

## Author Comment (AC4) · 28 Jul 2020

I am sorry to say that I made some technical mistakes in the first revised version of essd-2019-212. I double-checked the manuscript carefully and the hyperlink missed the last character 'c' was added. The other technical mistakes corrections such as spelling mistakes were corrected shown in a marked-up manuscript version.